# Influence of Storage Time on the DNA Integrity and Viability of Spermatozoa of the Spider Crab *Maja brachydactyla*

**DOI:** 10.3390/ani13223555

**Published:** 2023-11-17

**Authors:** Elba Rodríguez-Pena, Diego Suárez, Graciela Estévez-Pérez, Patricia Verísimo, Noelia Barreira, Luis Fernández, Ana González-Tizón, Andrés Martínez-Lage

**Affiliations:** 1CICA (Centro Interdisciplinar de Química e Bioloxía), University of A Coruña, 15071 A Coruña, Spain; elba.rodriguez@udc.es (E.R.-P.); ana.gonzalez.tizon@udc.es (A.G.-T.); 2Department of Computer Science, University of A Coruña, 15071 A Coruña, Spain; d.suarez@udc.es (D.S.); noelia.barreira@udc.es (N.B.); 3Department of Mathematics, University of A Coruña, 15071 A Coruña, Spain; graciela.estevez.perez@udc.es; 4Centro Oceanográfico de Santander (IEO-CSIC), 39004 Santander, Spain; patricia.verisimo@ieo.csic.es; 5CITIC (Research Center of Information and Communication Technologies), University of A Coruña, 15071 A Coruña, Spain; 6Department of Biology, University of A Coruña, 15071 A Coruña, Spain; luis.fernandezr@udc.es

**Keywords:** sperm storage time, sperm viability, *Maja brachydactyla*, comet assay, sperm degradation

## Abstract

**Simple Summary:**

On one hand, the females of some crabs can store the sperm from copulations for several years. On the other hand, it is reported that DNA repair mechanisms in sperm cells are less effective than in other cells. Taking into account these two considerations, we conducted a study into the viability and possible genetic damage of DNA in spermatozoa stored in female spider crabs for up to 14 months. The results show that during the first 3–4 months, both the viability and the DNA integrity fall considerably.

**Abstract:**

Natural populations of the spider crab *Maja brachydactyla* constitute a fishery resource of great economic importance in many countries. As in the rest of eubrachyurans, the females of this species have ventral-type seminal receptacles where they store sperm from copulations. Sperm can be stored in these structures for months and even years before egg fertilisation, with the consequent degradation of the sperm cells during the time. In this work, we analyse the viability and the possible genetic damage in sperm accumulated in the seminal receptacles of *M. brachydactyla* females as a function of the storage time (from 0 to 14 months) using the comet assay technique. On one hand, we developed an algorithm for comet image analysis that improves the comet segmentation compared with the free software Open comet v1.3.1 (97% vs. 76% of detection). In addition, our software allows the manual modification of the contours wrongly delimited via the automatic tool. On the other hand, our data show a sharp decline in sperm viability and DNA integrity in the first four months of storage, which could lead to a decrease in the fecundity rate and/or viability of the embryos or larvae from the second and third clutches of the annual cycle if the repair capacity in these gametic cells is low.

## 1. Introduction

Natural populations of the spider crab *Maja brachydactyla* constitute a fishery resource of great economic importance in many countries, such as the United Kingdom, Ireland, France, Spain, Portugal, and Morocco [1,2,3]. This species of decapod crustacean belongs to the Eubrachyura group, the females of which are characterised by having ventral-type seminal receptacles. These are structures in which females store sperm from copulations [3,4,5,6], which they later use to fertilise one or more broods [3,7,8,9]. The number of broods during an annual spawning period can vary according to the latitude, with three in Galicia (NW Spain) [3], two in France [1], and one or two in the English Channel [10]. In Galicia, a reproductive migration to deeper waters (30–100 m) takes place in early autumn, where mating occurs [3]. After mating (in January and February), females carry out a return migration to shallow waters for the incubation of the first brood of the annual cycle [11]). The incubation of each brood lasts about 1.5–2 months depending on the water temperature [3]. The waiting times between hatching and new spawning are about 3.4 days on average on the Atlantic coast of the Iberian Peninsula [3]. Thus, male sperm can be stored in the seminal receptacles of females for at least 6 months in primiparous females, and longer in multiparous females if they did not spend all the sperm from previous years.

In decapods, the sperm storage time after mating varies from hours to years and depends on the reproductive biology, the type of the female’s seminal receptacle, and environmental factors (i.e., water temperature) [12,13,14]. *Chionoecetes bairdi* females are capable of producing 100, 97, and 71% of fertilised eggs in the same year as mating, and one and two years after mating, respectively [15]. *Callinectes sapidus* and *Metacarcinus magister* females are capable of producing fertilised eggs using spermatophores stored between 7 and 11 months and 2.5 years, respectively [6,16,17].

The brachyuran spermatozoa are aflagellates, as in other decapods and many other crustaceans [18]. Their structure consists of a spherical acrosome with the cylindrical perforatorium located centrally, enclosed within the cell nucleus so that only the anterior part of the acrosome covered by the operculum protrudes from the nucleus [18,19,20]. Like other spermatozoa, they have a low capacity for DNA repair due to their limited antioxidant defenses. This implies that these cells are potentially more susceptible to genotoxins than oocytes and somatic cells [21,22,23,24], both at the nuclear and mitochondrial DNA levels [25,26], with oxidative stress being the main mechanism of DNA fragmentation [27,28]. As García-Rodríguez et al. reviewed in 2019 [29], there are three reasons why spermatozoa may present genetic damage: (i) defects in chromatin packaging [30], (ii) the incidence of apoptotic processes in mature spermatozoa, as apoptosis cannot be performed efficiently [31], and (iii) the incidence of genotoxic effects [32].

One of the reasons why repair systems are ineffective in spermatozoa is associated with the higher level of chromatin fibre compaction. In general, the typical chromatin’s histone proteins are replaced by protamines or protamine-like proteins. In the sperm of *Cancer* sp., the DNA is associated with histone-like proteins but with a smaller nucleosomal linker [33]. However, in *M. brachydactyla*, histones are present in sperm with lower proportions of histone H3 and higher proportions of histone H2B than other core histones [34]. Moreover, the histone H3 and H4 in sperm nuclei is acetylated, which is related to the decondensed state of *M. brachydactyla* sperm chromatin, which confers a lower state of chromatin condensation than in other spermatozoa [33,34,35,36].

The comet assay, also known as single-cell gel electrophoresis (SCGE), is a technique that combines DNA gel electrophoresis with fluorescence microscopy to understand the migration of DNA strands from damaged cells [37]. The method consists of embedding the cells to be studied in agarose, lysing them, and migrating the DNA via electrophoresis [38]. Undamaged DNA remains intact, without any migration, whereas fragmented DNA will migrate towards the anode and acquire a shape similar to the tail of a comet. This shape is visualised using fluorescent staining and microscopic observation of the slides.

This technique has a wide variety of applications in the study of genetic damage generated by genotoxic products [39,40,41,42], but it is also useful for determining natural genetic damage in spermatozoa [43,44,45]. For this reason, it is commonly used in fertility clinics and in livestock reproduction. In crustaceans, its use is less widespread, and most studies focus on determining genetic damage caused by genotoxic products. In this way, the comet assay has been used to detect genetic damage in the gill, hepatopancreas, and haemolymph in *Litopenaneus vannamei* [46], the haemolymph in *Gammarus fossarum* [47], and the spermatozoa in *Macrobrachium rosenbergii* [48].

To our knowledge, this technique has never been used to detect natural genetic damage in crustacean sperm. Only the integrity of sperm has been assessed based on the morphology of the plasma membrane and acrosome [49]. In this work, we analysed possible genetic damage in spermatozoa accumulated in the seminal receptacles of *M. brachydactyla* females as a function of storage time. If the repair capacity in these gametic cells is low, the associated sperm damage in the second or third broods of the annual cycle will result in a lower fecundity rate and/or viability in the embryos and larvae.

## 2. Materials and Methods

### 2.1. Capture of Test Specimens and Sample Collection

Divers from Aquarium Finisterrae of A Coruña captured 38 females and 26 males of *M. brachydactyla* in the Artabrian Gulf. The females were dissected in different batches in order to analyse the effect of storage time on sperm quality. Sperm storage time in the different study females ranged from 0 to 14 months (see Section 3.2 (Results section) for further details), covering the periods corresponding to the first, second, and third broods of a breeding cycle, as well as the first brood of the following annual cycle.

The females used for this experiment belong to three groups with different characteristics. The first is composed of all the females of 0, 1, 2, and 3 months of sperm storage time, which performed their terminal moult in captivity. Each of these females mated with a single male in controlled conditions. The second group consisted of females that had been in captivity for at least one year in the absence of males, and which had not laid eggs in the last breeding cycle or had laid them unfertilised. This fact was considered as an indication that the females no longer carried sperm in their seminal receptacles. Furthermore, some of the females in this set were dissected to verify that their seminal receptacles were empty. The rest of these females were crossed in captivity with a single male and used for the experiment described in this article. Finally, the third and least numerous group of females consisted of specimens carrying sperm from copulations in the natural environment, and the storage time was estimated based on the mating season reported in the literature for this species (late autumn, in [3]). We have been forced to use these three types of females due to the difficulty in obtaining virgin females of *M. brachydactyla*. All the females were kept in absence of males from the time of mating in captivity or their arrival at the Aquarium Finisterrae (in the case of the third stock) until their dissection. They were kept in a semi-open circuit tank under the temperature and salinity conditions of the spring months in the Artabrian Gulf (temperature: 13 ± 1 °C, salinity: 35 g/L).

Females were anaesthetised in a freezer at −20 °C for 15 min before dissection. Then, the seminal receptacles of each female were removed, and their membranes cut to collect the sperm masses contained inside. Of the 38 females dissected, 8 were discarded because the amount of stored sperm was very low and did not reach the minimum number of spermatozoa required for the results to be significant. Six females were dissected at 0 months (less than 24 h from copulation) and they served as controls: negative control, in the case of no treated spermatozoa, and positive control, in the case of spermatozoa exposed for 2 h to H_2_O_2_ 10 mM in darkness. Sperm quality was evaluated in terms of viability and DNA integrity.

### 2.2. Viability Test

The contents of each right seminal receptacle were homogenised via vortex for 10 s at 2000 rpm in 20 mL of filtered (0.45 µm) and sterilised seawater. From this cell suspension, 100 µL was taken, and a 1:50 (*v*/*v*) dilution in seawater and a preliminary sperm count were performed in a Neubauer chamber. From this count, the suspension was diluted to a concentration of approximately 1000 cells/µL and stained with fluorescein diacetate (FDA) at a final concentration of 0.2 ng/µL. This fluorochrome indicates the presence of esterase activity and cell membrane integrity, both of which can be used as indicators of cell viability [50]. From this point on, the whole process was carried out in darkness. After a 15 min incubation at room temperature, each cell suspension was prepared and visualised using a Nikon Microphot-FXA microscope. The percentage of sperm viability was estimated as the proportion of cells emitting fluorescence out of a total of 150 cells examined for each female.

### 2.3. Comet Assay

The procedure used was that described by Olive et al. [37], with slight modifications. Two slides were prepared from the seminal mass of the left receptacle of each dissected specimen. Four tests were conducted on different days, with at least one positive and one negative control each. Microscope slides were pre-coated with a base layer of 130 μL of 1% (p/v) normal-melting-point agarose. The cells were resuspended in 0.7% low-melting-point agarose in phosphate-buffered saline. Subsequently, 100 μL of the cell suspension (approximately 6 × 10^6^ cells) was dropped onto each of the pre-coated slides, and a coverslip was placed to spread the preparation, which was left to gel at 4 °C. After solidification of the agarose, slides were covered with 80 μL of 0.7% (p/v) low-melting agarose and then placed in lysis solution (2.5 M ClNa, 100 mM Na2EDTA, 10 mM Tris-HCl, 1% (*v*/*v*) Triton X-100, 10% (*v*/*v*) DMSO) at room temperature for 10 min in the dark. All the following steps of the comet assay were performed in darkness. After lysis, the slides were transferred to an electrophoresis chamber containing an alkaline solution (300 mM NaOH, 1 mM Na_2_EDTA, pH > 12) for 30 min at 4 °C for DNA unwinding. Electrophoresis was performed under 0.66 V/cm and 300 mA for 20 min at 4 °C. The slides were removed from the chamber and washed three times with a neutralising buffer (0.4 M Tris-HCl, pH 7.5) for 5 min at 4 °C. Then, the preparations were stained with SYBR Green I nucleic acid gel stain and were visualised and photographed using a Nikon Microphot-FXA microscope equipped with the NIS-Elements D 3.10 software and a digital camera DS-Qi1Mc. For each slide, approximately 100 photographs were taken for subsequent analysis.

### 2.4. Automatic Comet Segmentation

The automatic segmentation of the comets is a challenging task due to the high variability of the comet assay images. There are some tools in the literature to deal with this problem. On the one hand, Comet Assay IV [51] and Komet 7 [52] are two popular commercial tools for the automatic segmentation of comet assay images. On the other hand, open source tools such as Open Comet [53] have been widely used for this task, mainly for research purposes. Initially, we used the free tool Open Comet to analyse our images. However, this plugin was not able to correctly segment the cells due to the lightness conditions and the comet features of our dataset. For this reason, we have developed a computer vision algorithm to automatise the segmentation task. This algorithm consists of four main stages: preprocessing, candidate comet location, head segmentation, and tail segmentation [54] or, on request, AML.

The resulting algorithm was applied to comet photographs to obtain several parameters related to DNA integrity.

### 2.5. Statistical Analysis

For the study variables, each female was considered as an experimental unit. Therefore, the data used were the means within each female.

To explore the linear relationship of the sperm storage time in seminal receptacles with both the percentage of sperm viability and variables representative of DNA degradation obtained via the comet assay, Spearman’s rank correlation coefficients were calculated. In addition, segmented linear regression (SLR) models were applied to fit such relationships. Generally speaking, SRL allowed us to obtain a better interpretation of the temporal evolution of both viability and DNA integrity, when compared to other standard methods widely used in the literature. We have also applied the simple linear regression model for raw data and for log-transformed data, although the relationships did not appear linear.

Segmented or broken-line regression models allow us to fit piecewise linear relationships between a response variable and one or more explanatory variables, represented by two or more straight lines connected at unknown points. These models are a common tool in situations where it is of interest to assess threshold values where the effect of the covariate changes. They provide a valuable interpretation in terms of the changepoints and the slopes [55,56].

The distribution evolution of genetic damage versus storage time was analised using a ridgeline plot. A ridgeline plot [57] shows the distribution of a numeric variable for several groups through histograms or density functions aligned on the same horizontal scale and presented with a slight overlap.

Statistical analyses were conducted using R version 4.1.0 [58] in RStudio version 1.4.1717 [59], with the following packages: segmented [60] and ggridge [61]. Statistical significance was set at *p* < 0.05.

## 3. Results

### 3.1. Optimisation of Automatic Comet Segmentation

We have developed a computer vision algorithm to improve the comet segmentation task. In the preprocessing stage, morphological filters reduce the noise in the images. Since the comets are brighter than the background, a set of candidate regions are located using thresholding algorithms. Within these regions, the heads of the comets are the brightest spots, so they are identified using thresholding algorithms and shape constraints (size, convexity, and circularity). Finally, morphological filters smooth the candidate regions, and the tails are delimited via computing the convex hull of the region. Nevertheless, since some cells could not be degraded, the comets could have no tail. In these cases, the candidate regions contain only the head and maybe some background around it. In order to identify both types of cells, a decision tree classifier was trained using four features: the ratio of head to tail, the head location with respect to the tail, and the average intensities of the head and tail pixels. Moreover, comets with an inappropriate ratio of head to tail or not parallel to the *x*-axis were discarded.

The proposed algorithm was tested on a set of 13 images (around 8–10 cells per image) with manual ground truth, and its results were compared to the Open Comet algorithm [46]. The Intersection over Union (IoU) metric (Intersection over Union or Jaccard index is the size of the intersection divided by the size of the union of two sample sets. In our case, the sample sets are the pixels marked manually by an expert as part of the cell comet and the pixels computed via the algorithm) was computed, and its results are shown in Table 1. Moreover, the Open Comet algorithm was able to locate 76% of the cells, whereas our approach found 97.33% of them. Clearly, our algorithm outperforms both comet location and segmentation on this particular dataset.

This algorithm was implemented in Python using the OpenCV and Sklearn libraries and was integrated into a GTK application. This application has functionalities such as opening an image file, creating a project, applying the segmentation algorithm, or saving the statistical results computed in the image. Moreover, the application provides an editing tool to manually segment the comets in the images, correct the wrongly detected contours, or delete segmented artefacts.

### 3.2. Sperm Viability and DNA Integrity Analysis

Firstly, the data for the study variables were subjected to an exploratory analysis. Then, the above-mentioned regression models were fitted to explain the percentage of sperm viability versus sperm storage time in the seminal receptacles (in months). DNA integrity was assessed through the tail moment parameter measured via our computer vision algorithm. This parameter is defined as the product of the tail length and the fraction of total DNA in the tail. The higher the value of this parameter, the greater the DNA degradation. Similarly, for viability, the regression models of tail moment versus sperm storage time in seminal receptacles (in months) were fitted. In both cases, the segmented linear regression model with a cutoff point provided the best fit to our data.

Regarding viability, the percentages for the 30 females tested are shown in Table 2.

In Figure 1, it can be observed that there is no direct proportionality (linear) between sperm viability and storage time, but there is a significant monotone decrease, which is confirmed via the Spearman correlation coefficient (r = −0.9109; *p* = 2.808 × 10^−12^). In addition, the fitted SLR model shows that sperm viability declines sharply between the first and fourth month of storage (at a rate of 13.30% per month), from a maximum value of 100% to around 40%. From the fourth month (estimated breakpoint at 4.22 months), the slope of the regression line becomes less steep (a decrease of 4.60% for each completed month). The viability continues to decrease, but at a slower rate, reaching 0% viability in the next 10 months. The regression analysis showed a significant adjusted determination coefficient R^2^ = 0.7655 (*p* = 8.8741 × 10^−14^).

As for DNA integrity (Figure 1), a significant monotone increase is observed between tail moment and storage time. This is confirmed using Spearman’s correlation coefficient (r = 0.7324; *p* = 4.212 × 10^−6^). The fitted SLR model shows that the tail moment value drops sharply during the first three months of sperm storage (at a rate of 24.24 units per month), from values close to 0 to values close to 68%. However, after the third month (estimated breakpoint at 3.17 months), the DNA degradation stabilises (estimate slope: 0.21), with tail moment values around 70%. The regression line goes from being significant in the first section (95% CI for slope [17.5490, 30.9370]) to being non-significant in the second (95% CI for slope [−2.5757, 2.1576]). The coefficient of determination of the segmented regression model for this case is R^2^ = 0.8143 (*p* = 2.6581 × 10^−12^).

In order to observe more general characteristics of tail moment distribution, a ridgeline plot has been obtained, as shown in Figure 2. This plot, which displays the size density distribution of tail moment for the different storage times, confirms the negative impact of storage time on sperm quality: a waste of normality, an increase in spread, and the onset of females showing atypical performance. All this can be interpreted as cells with little degraded DNA coexisting with cells with highly degraded DNA over the same storage time.

## 4. Discussion

The experimental design of this work has been influenced by the difficulty of obtaining virgin females to cover the whole range of sperm storage times. Virgin females with sperm storage times from 0 to 3 months are the ideal specimens to study sperm degradation as a function of storage time. In this storage time interval, a strong degradation of the stored sperm is already observed, especially in terms of DNA integrity. The rest of the females used may have some disadvantages.

For the females that carried sperm from copulations in the natural environment, the time since copulation was estimated in an approximate way, taking as a reference the mating season of this species in the Artabrian Gulf. Thus, we assumed a small error in the storage time of approximately ±1 month in these seven females. All of them carried sperm stored for at least 11 months, so the assumed error in these cases is less than 10%. In addition, these females could have copulated with several males. Although the time between copulations would have been short compared to the sperm storage time, sperm cells with different stages of degradation could be observed. These differences would not be due so much to the different storage time in the receptacles as to inter-individual differences between the sperm qualities of the males due, for example, to their age.

Moreover, inter-individual differences could also exist between females in terms of the effectiveness of DNA repair mechanisms. Although females older than two years as adults have been excluded from this experiment, there could be small differences in this respect between recently matured and multiparous females. We are aware that these factors could have influenced the results of our experiment. However, we have tried to minimise their effect as much as possible in both the experimental design and the analyses.

Our results show that sperm deterioration increases continuously with the storage time (Figure 1), following the “sperm aging decelerates” model proposed by Reinghardt [62]. Previous studies have already shown how the age of the sperm has a negative effect on its quality [63,64,65]. We have to take into account that it is unusual for ejaculated sperm to survive for more than 5–6 days in mammalians or 12–13 days in birds [66]. However, this is more variable in other taxa. For example, in insects, this time can range from 7–10 days in the butterflies *Euphydryas editha* or *Papilio polyxenes* [67] to more than 30 days in *Drosophila melanogaster* [68]. In amphibians, the interval is even longer, ranging from twelve days in *Ambystoma mexicanum* [69] to one year in *Salamandra salamandra* [70].

In this sense, our results for *M. brachydactyla* show that there is variability in the deterioration of the spermatozoa in every female, so both damaged and undamaged sperm are visible simultaneously for each individual (except for the control females). For instance, the 5.2-month sperm exhibits two peaks in its tail moment distribution, one with values of 50 and another with values of around 75. As for the sperm stored for 11.6 months, it shows some spermatozoa with little genetic damage (a peak of tail moment at 20), whereas the other cells are very deteriorated (a range of tail moment between 40 and 120) (Figure 2).

The existence of genetic damage in ejaculated sperm has been mentioned for other crustaceans. Thus, Lacaze et al. [24] detected it from four days after ejaculation in *Gammarus fossarum*, including a 5% damage in the DNA within the tail of the comet from the control individuals [47]. Rani et al. [48] found that in *Macrobrachium rosenbergii*, the percentage of genetic damage in sperm in the control individuals is around 20%, while Erraud et al. [71,72] found very high values of genetic damage in the *Palaemon longirostris* prawns obtained on the Seine estuary (naturally exposed to environmental contamination). On the contrary, Feng et al. [49] observed only little sperm fragmentation in the *Penaeus monodon* control individuals.

A parameter closely related to the survival ability of spermatozoa is the acrosome reaction [14]. Many factors are believed to induce acrosome reaction in vitro or in vivo in decapod spermatozoa. The egg water, calcium ionophore, pH, concentrations of various ions, cold shock, sperm maturity, bacterial concentration in the female sperm storage organs, male size, male mating history, season, and pressure affect the induction of acrosome reactions in decapods [73,74,75,76].

It is considered that mature spermatozoa are incapable of repairing the DNA damage due to the extreme compaction of their DNA and to their limited transcriptional ability [29,77]. In addition, the cytoplasm of *M. brachydactyla* spermatozoa occupies a minimal surface area [20], so the number of ribosomes should be low and, therefore, have decreased translational activity. This implies that the enzymes involved in repair mechanisms will be present at a basal level, which would explain the DNA fragmentation visualised as comet tail. However, the spermatozoa with fragmented DNA still have fertility and development potential, since the oocyte repair machine can be enough to repair the DNA damage [78]. Thus, it is considered that, in humans, viabilities over 50% are evidence of fertility [79]. Therefore, a high damage and/or a low viability of sperm does not necessarily imply a drastic reduction in larval hatching. In addition, as mentioned previously, chromatin in spider crabs has a lower compaction degree [34], so the repair mechanisms could be a little more effective than in other spermatozoa. Another characteristic of crustacean spermatozoa is that, despite being aflagellate, they can achieve short-term motility through an abrupt eversion of the acrosome. Morrow [80] proposed that aflagellated spermatozoa evolved from monandric mating systems, in which the absence of spermatic competition can relax the selection pressure over the mates to produce motile spermatozoa.

Seminal receptacles are common in several taxa without motile sperm. These structures comprise an evolutionary solution that allows females to lay more than one brood without the need to copulate again. Its existence is very widespread among the Myriapoda [81,82], worms [83], Chelicerata [84,85], Tardigrada [86,87], and Onychophora [88,89] and Insecta [90,91]. It must be considered that the moment of copulation tends to be one of the moments when the individuals are more defenceless against predators; therefore, the time is usually minimised. Although spider crabs have few predators (octopi, sharks, morays, congers...) because of their external structure, and they mate when their carapace is already hard, many other species with seminal receptacles have many predators and/or copulate when their shell is soft.

Migration to deeper waters for copulative events occurs in several species of crab [92,93]. In the northwest Iberian Peninsula, individuals of *M. brachydactyla* migrate to deeper waters to mate after completing the terminal moult [3,94,95]. At this latitude, females can lay up to three broods. The first broods contain less and bigger eggs, while the next broods contain more and smaller eggs [96]. Although this has been explained according to the available food [97,98], it could also be interpreted as a compensation mechanism for the genetic damage observed in this work. It has been confirmed in laboratory studies that the number of surviving larvae decreases throughout their different stages, with a drastic reduction in survival from the moulting to megalopa stage [99,100,101]. An increase in the quantity of eggs could compensate for the decrease in offspring survival.

There are disagreements over the possible consequences of storing sperm in terms of the fertility and viability of offspring in crustaceans. Thus, it was observed in *Pseudocarcinus gigas* that the sperm stays viable for at least four years, hatching larvae without, apparently, any disorder [102] and, on the contrary, the stored sperm from *Chionoectes bairdi* has a limited life cycle and it is not viable after two years [15]. As well as the possible influence of stress caused by the laboratory conditions, these differences could be related to the better ability to repair DNA in some species or individuals than in others.

Simeó et al. [103] compared the number of larvae produced by *M. brachydactyla* females kept in captivity in the absence and presence of males. They found that larval production was higher in tanks where males were present. It could be interpreted as this species having the ability of mating throughout the whole year, and these migrations for mating in autumn are only used to facilitate the exchange of gametes. These mating congregations also favour the multiple paternity (polyandry), a reproductive mechanism described for this species [104]. We verified, in some dissections of seminal receptacles, the existence of separated ejaculates. In these cases, the existence of differentiated sperm masses could be related to the time between copulations being several months and to the female not having used the entirety of the first ejaculate. However, in laboratory experiments, when a female mated with two males within a few days, no distinct sperm masses were observed inside the seminal receptacles. This could indicate a higher mixing of male sperm when the time between copulations is short and, therefore, a higher probability of multiple paternity of the broods. When multiple males copulate with a female, the offspring will benefit from the genetic variability, very valuable in an environmental-change scenario, as it provides more variability for natural selection [105], including inter-individual variability in DNA repair ability.

The existence of seminal receptacles in spider crabs as structures that allow sperm storage facilitates the spermatic competition associated with multiple paternity. This competition is possible because females can mate with several males, and they can store viable sperm for long periods of time [106]. This competition can be very strong because only part of the sperm accumulated in a full seminal receptacle will be used to fertilise the brood (15% in *Halicarcinus cookie* [107]). The polyandry can be found in a spontaneous way or owing to the necessity for females to fill their receptacles completely. It may happen that a large female did not fill her seminal receptacles entirely with one copulation, needing several mates to do so. This could be due not only to the difference of size between the male and the female but also to males not employing the entirety of their sperm in a single copulation with the purpose of looking for mating with other females, as described in the snow crab *Chionoecetes opilio* [108]. Females of different species can package their yolk into eggs of different sizes and males can package the sperm into spermatophores of different sizes. In the case of males, they can adjust the ejaculation size according to female numbers (sex ratio) or the situation (e.g., the presence of male competitors) to maximise their reproductive success [109]. Moreover, in *C. opilio*, it has been reported that the weight of the ejaculate decreases as the males grow old [110]. Another factor that could cause the incomplete filling of the seminal receptacles in spider crabs is that, after the terminal moult and reproductive migration, males are not physiologically recovered, so they may ejaculate a smaller amount of sperm. In addition to detecting these physiological deficiencies after migration, Corgos et al. [95] reported that the first males to migrate, and thus to mate with females, are the smallest. This strategy allows smaller males to maximise their mating opportunities, but it could lead females to copulate several times to fill their receptacles completely.

The results found in this study suggest a basic question: if there is such a strong deterioration in spermatic DNA, why is it stored? On this matter, McLay and López Greco [106] proposed a hypothesis about the origin of seminal receptacles in Eubrachyura. Males would deposit spermatophores that would adhere around the gonopore of the female, which would facilitate the evolution of an open receptacle created via the invagination of the genital area, which would offer better protection to the spermatophores. This evolutionary solution, which allowed the storage of sperm for long periods of time, resulted in increased genetic damage. However, it could also be related to processes such as the increase in mutation rates and/or the reduction in spermatic motility produced in species with multiple mating between both genders and a high production of sperm [62,64,65,110,111].

Although it has been suggested that the function of the seminal receptacles is to accumulate sperm to fertilise several clutches without the need for new copulations, this explanation does not agree with the data on sperm degradation over a few months obtained in this work. Moreover, this type of structure also appears in populations with a single annual brood [112,113]. On the other hand, the existence of mechanisms for the elimination of old sperm in the seminal receptacles of other majoids has been suggested [114], which also does not fit the hypothesis of seminal receptacles as long-term sperm-storage organs. The protection of sperm until fertilisation, the accumulation of sufficient ejaculates to fertilise all eggs in a brood, and the possibility of sperm competition and multiple paternity provided by these structures are explanations that better fit the results of this study.

In order to know the extent of the detected genetic damage, it is necessary to study its real implications and the capacity of repair mechanisms in brachyurans by conducting similar studies in species that show very different sperm viabilities, such as *Pseudocarcinus gigas* and *Chionoectes bairdi*. Inter-individual variability studies should also be carried out to assess the range of this phenomenon.

Regarding *M. bracydactyla*, it would be interesting to further investigate the causes and mechanisms of sperm degradation. In future studies, it will be necessary to evaluate other indicators of sperm quality, such as sperm activity, physiological indicators, and the expression of sperm-related genes. Nevertheless, we believe that this work provides an important starting point to address these issues in the future and raises new questions about the true function of seminal receptacles in crustaceans.

## 5. Conclusions

Firstly, the software developed by the authors and presented in this work improves the comet segmentation compared to the free software Open comet (97% vs. 76% of detection). In addition, it allows the manual modification of the contours wrongly delimited via the software. Although the development of this algorithm was not one of the main objectives of this work, but was intended to provide a free tool for our analyses, we believe it can be an effective and useful open resource for other researchers. Secondly, genetic damage occurs, and the viability of the sperm stored in the seminal receptacles of spider crabs declines, after 3–4 months. However, repair systems are apparently capable of repairing genetic damage in sperm in many cases, as other studies in this species have reported that hatchings still occur after one year of storage.

## Figures and Tables

**Figure 1 animals-13-03555-f001:**
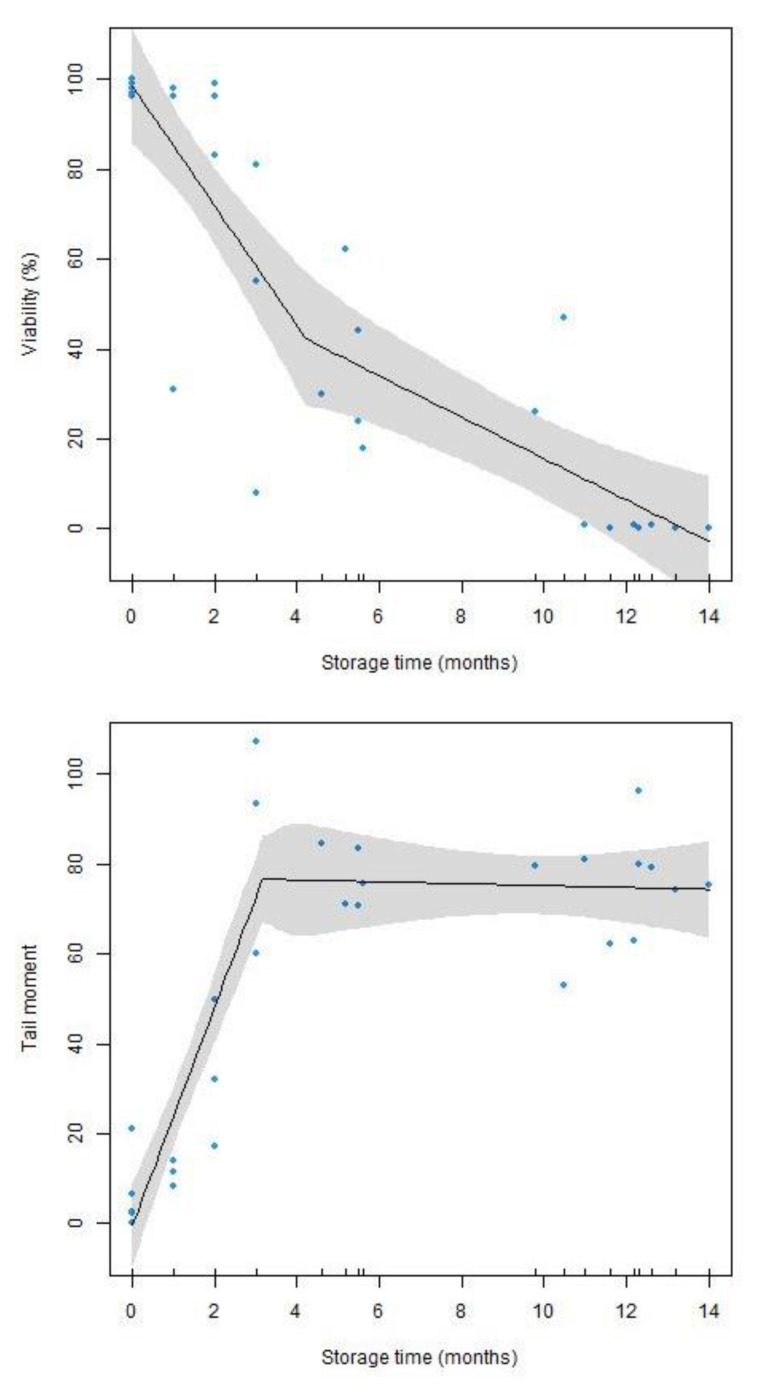
Segmented regression plots for sperm viability (at the top) and tail moment (at the bottom) versus storage time in seminal receptacles (months) with the shaded regions denoting 95% confidence intervals (R^2^ = 0.7655; *p* = 8.8741 × 10^−14^ for viability and R^2^ = 0.8143; *p* = 2.65811 × 10^−12^ for tail moment).

**Figure 2 animals-13-03555-f002:**
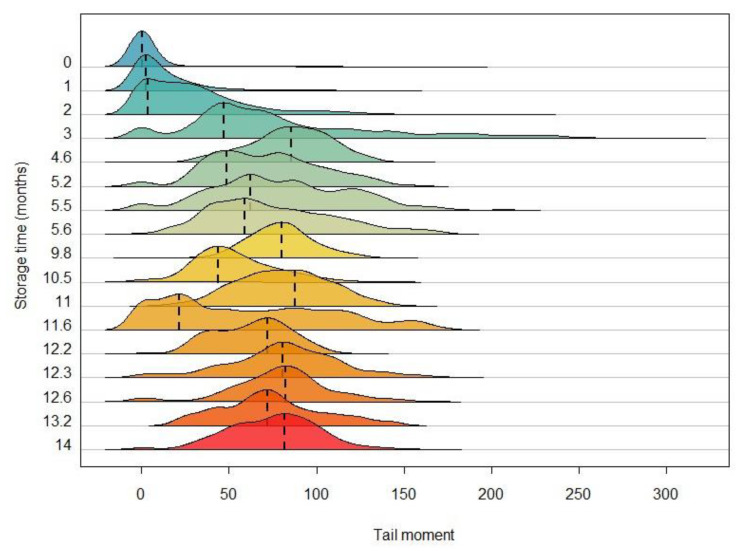
Ridgeline plot showing the size density distribution of tail moment as a function of storage time. Each curve represents the distribution of tail moment for the comets analysed for each storage time and their modal values are represented by dashed lines. The colour gradient from blue to red indicates an increasing gradient in sperm storage time.

**Table 1 animals-13-03555-t001:** The results of the segmentation algorithms in our dataset.

	Open Comet	Our Approach
IoUhead	0.43	0.71
IoUtail	0.30	0.70

**Table 2 animals-13-03555-t002:** The trial number, storage time, and percentage of viability of the female spider crabs used in this study.

Individual	Test No	Storage Time (Months)	Viability (%)
fem-3-1	trial-3	0	100
fem-4-1	trial-4	0	100
fem-1-4	trial-1	0	99
fem-1-2	trial-1	0	98
fem-1-3	trial-1	0	97
fem-1-1	trial-1	0	96
fem-4-2	trial-4	1.0	98
fem-6-2	trial-6	1.0	96
fem-6-1	trial-6	1.0	31
fem-5-1	trial-5	2.0	99
fem-6-3	trial-6	2.0	96
fem-4-3	trial-4	2.0	83
fem-6-5	trial-6	3.0	81
fem-6-6	trial-6	3.0	55
fem-6-4	trial-6	3.0	8
fem-3-2	trial-3	4.6	30
fem-3-3	trial-3	5.2	62
fem-3-4	trial-3	5.5	44
fem-3-5	trial-3	5.5	24
fem-3-6	trial-3	5.6	18
fem-4-4	trial-4	9.8	26
fem-2-1	trial-2	10.5	47
fem-2-2	trial-2	11.0	1
fem-3-7	trial-3	11.6	0
fem-2-3	trial-2	12.2	1
fem-2-4	trial-2	12.3	0
fem-5-2	trial-5	12.3	0
fem-1-5	trial-1	12.6	1
fem-5-3	trial-5	13.2	0
fem-2-5	trial-2	14.0	0

## Data Availability

The GTK application for comet segmentation developed in this work is available at https://github.com/dsuarezgarcia/freecomet. The rest of the data presented in this study are available on request from the corresponding author.

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
