# Peer review of "Influence of Storage Time on the DNA Integrity and Viability of Spermatozoa of the Spider Crab Maja brachydactyla"

_animals, 2023, doi:10.3390/ani13223555_

Round 1
Reviewer 1 Report (Previous Reviewer 1)
Comments and Suggestions for Authors
I have no further questions.
Author Response
We would like to thank the reviewer for taking the time to review our manuscript.
Reviewer 2 Report (Previous Reviewer 2)
Comments and Suggestions for Authors
The authors have made sufficient revisions and have addressed all the concerns. Therefore, the manuscript can be accepted for publication after a minor revision. There is room for improvement in both the introduction and the discussion sections of the manuscript. Several related articles have been overlooked. I have compiled a list of some relevant review articles for your reference. For instance, the review article titled "Molecular and Cellular Biology of the Crayfish Spermatozoon: Toward the Development of Artificial Reproduction in Aquaculture" in Reviews in Fisheries Science & Aquaculture (2019) contains extensive information about the integrity, viability, and storage time of spermatozoa in various crustaceans. You can find valuable insights regarding sperm storage duration in this paper.
Author Response
Please see the attachment

Reviewer 3 Report (Previous Reviewer 4)
Comments and Suggestions for Authors
I have reviewed the manuscript after the authors made major changes. The paper has substantially improved and the English is now in a very well readable and understandable form. I recommend the paper for publication now after few minor corrections are made (see belwo). Congratulation!
Recommended changes/corrections:
Line 101: replace “we want to analyse” with “analysed”
Line 361: replace “this” with “it”
Line 369 – 373: Delete “Thus” and brake sentence into two.
Line 393: replace “on the” with “of”
Line 413: replace “until” with “up to”
Line 430: Replace “along” with “throughout”
Line 436: replace “in experiments in laboratory” with laboratory experiments”
Line 441: replace “as provides” with as it provides”
Line 451: move “entirely” behind “receptacles”
Line 452: Insert “not” after “This could”
Line 460: replace “ejaculated” with “ejaculate”
Line 474: delete “deposited in it
Line 490: replace “the real implication of it” with “its real implication”
Line 491: delete “to this”
In addition: Please check the use of the term “terminal moult”. Terminal moult is very rare in crustaceans and I am not aware that it exists in Maja.
Comments on the Quality of English LanguageThe quality has substantially improved and is now on an acceptable level for the journal.
Author Response
Please see the attachment

This manuscript is a resubmission of an earlier submission. The following is a list of the peer review reports and author responses from that submission.
Round 1
Reviewer 1 Report
Comments and Suggestions for Authors
In the present study, authors indicated that a sharp decline in sperm viability and DNA integrity in the first four months of storage through performing the Comet assay. However, in my opinion that the experimental design is too simple. Authors did not analyse that why and how the sperms will be degraded during that period. In addition, considering that a computer vision algorithm developed is the innovation of this study, the manuscript is more suitable to be published as a methodological paper, after highlighting this part. Thus, I suggested rejection for the manuscript.
The abstract is too simple. Authors did not summary all of their results in the present study. The abstract is suggested to be re-written.
Authors mentioned in the Introduction that “This species of decapod crustacean belongs to the Eubrachyura group, whose females are characterised by having ventral type seminal receptacles. These are structures in which females store sperm from copulations, which they later use to fertilise one or more broods. It is not clear that how long for one brood of Maja brachydactyla. According to the author’s results, a sharp decline in sperm viability and DNA integrity in the first four months of storage. Whether your data can support the phenomenon that the sperm stored in ventral type seminal receptacles can used for more than one brood.
The experimental design is too simple. Authors only determined that a sharp decline in sperm viability and DNA integrity in the first four months of storage. But why and how the sperms will be degraded is still not clear. In the Introduction (Line 68-76), previous studies have been provided some possible reasons for the sperm degradation. Authors should have provided some possible mechanisms that results in the sperm degradation.
As a new vision algorithm developed based on machine learning, the authors should add relevant details, such as train set size, and how the features are selected. And only 13 images might be not sufficient for a test set.
In the results section, some descriptions should be appeared the materials and methods section, like line 228-230.
Comments on the Quality of English Language
Minor editing of English language required
Reviewer 2 Report
Comments and Suggestions for Authors
In the manuscript entitled “Influence of storage time on the DNA integrity and viability of spermatozoa of the spider crab Maja brachydactyla” the authors design an experiment to investigate the effect of sperm storage (a period of 0 to 14 months) in the female seminal receptacles on the DNA integrity and viability spermatozoa in Maja brachydactyla. The idea of this experiment is very good and can be very useful for crab aquaculture.
However, there is a big problem with the sampling. All females of 0, 1, 2, and 3 months undergo mating in captivity (laboratory situation). But the samples for 4-14 months were captured from nature. So basically, some data are from the females that undergo mating at the lab and some data from the females from nature. The author mixed the data and suggest that after 4 months the sperm quality goes down. The reduction in the sperm DNA integrity and viability of spermatozoa can be because of having diffident samples from different sources and not because of the storage time. The reduction in sperm quality exactly happened for the samples after 4 months (the samples which were obtained from nature).
Since in the female seminal receptacles, the sperm are directly in interaction with the water. Environmental factors such as different salinity, water temperature, calcium in the water, etc can affect sperm quality. Therefore, the results and conclusion of this study are not reliable.
Moreover, for the samples from nature, the authors estimated the sperm storage time based on the mating season reported in the literature. This method is not reliable. Better to mate all crabs at the lab to make sure about the time and the environmental parameters.
In addition, there is also a problem with almost all sentences in the manuscript. The manuscript must be revised by a native English speaker.
Therefore, I think the present manuscript is not suitable for publication in “Animals”
My detailed comments are below:
There are many keywords. Is this according to the journal guideline
Introduction
Line 51: 3.4 days? Or 3-4 days?
Line 73-74: What H2B, H3, and H4 stand for? First give the full name.
Materials and methods
Line 102: were did you capture males? Capture only females? Didn't the crabs do mating at the lab? How can you make sure about the time of sperm storage if you capture females from nature?
Line 110-111: estimation of the sperm storage time based on the mating season reported in the literature seems to be not a reliable method. Better to mate all crabs at the lab. Do you have any solid evidence for the estimation method?
Moreover, the source of all samples must be the same. The sperm stored in the natural environment was affected by different environmental factors from the lab samples. For example, different salinity, water temperature, etc. Also, the sperm was delivered from different males living in different situations.
The decline in sperm quality in Fig 2 after 3-4 months. This can be because of having diffident samples from different sources and not because of the storage time.
Comments on the Quality of English LanguageThe manuscript must be revised by a native English speaker.
Reviewer 3 Report
Comments and Suggestions for Authors
The manuscript investigate the viability and possible genetic damage of DNA in spermatozoa stored in female spider crabs for up to 14 months, The results showed a sharp decline in sperm viability and DNA integrity in the first three-four months of storage. This work are well conducted and well wrote. However, several sentences need revision. For example, some paragraph such as 4-6 in introduction section, the methods can be described more refine.
In line 109, The rest of the females carried sperm from copulations in the natural environment in their seminal receptacles. If a female carried sperms from several males while the experimental females cross with a single male.
Figure 1 needs to be processed.
In all, the authors need design several experiments to improve the results, for example detect sperm activity, physiological indicators and expressions of sperm-related genes.
Reviewer 4 Report
Comments and Suggestions for Authors
The manuscript provides important new information on the mating process of a commercially important crab. This is of interest for a range of scientists, for example in fisheries resource management, aquaculture and general crustacean science. The comet assay was used to assess crustacean sperm damage for the first time. Methodology and quality of data are adequate. However, it is obvious that the manuscript is not written by a native English speaker, or someone who speaks it scientifically. It often suffers from the incorrect use English. This makes understanding sometimes difficult. A substantial part of my comments is on the English (see below in detail). Interestingly, Introduction, M&M and (partially) Results are exempt from this and seem to have been written by a different person than other parts, especially Discussion. I therefore recommend this manuscript to be considered for publication only after revision. This includes a check of the English by a competent person.
Please find my detailed advice below.
Abstract:
Line 31: replace “during” with “for”
Materials and Methods:
Line 114: At what temperature were females frozen and in how (water ice, liquid N2 etc.)?
Lines 120 – 153: Are all these methods developed by the authors or do they follow xxx from other researchers? This should be clearly stated and, if necessary, referenced.
Results
Fig. 1. Delete this figure. It is not really necessary.
Table 1. Replace “Our” with “Present”
Line231: Replace “showed” with “shown”
Line 253: rephrase
Line 260: “For both….” This belongs into M&M, too.
Line 271: Replace “smoother” with “less steep” or “flatter”
Line 278: Rephrase sentence or break up
Figures 2 – 5 are interlinked. At least Figs 3 – 5 depict the same thing in different ways. This should be reduced/condensed, for example in placing 3 – 5 as panels A,B,C into a single figure. However, it would be better to reduce the overall depiction of tail moment vs storage time.
Discussion:
Lots of small inaccuracies of the English, in the first para, for example, line 316 “showed” (use shown), line 323 “in” other crustaceans (use for), line 327 “find” (use found),
Line 317: Change to “is visible simultaneously….”
Line 339: Rephrase completely and remove “, but despite the absence of flagellum”
Line 370: Replace “interchange” with “exchange”
Some sentences are too long and winding, for example those starting in lines 375 and 377. Break up and rephrase.
Line 381: Rephrase “which facilitates the female when laying eggs”
Line 383: Replace “is very spreaded”. This is not English
Line 402: Delete “production” 2x
Conclusions:
Line 446: replace “restoring” by “repairing”
General:
The term “copula” is a statistical term and should be replaced by “copulation” throughout the ms
Use of expressions like “On the one hand” and “Thus”
Comments on the Quality of English LanguageIt is obvious that the manuscript is not written by a native English speaker, or someone who speaks it scientifically. It often suffers from the incorrect use English. This makes understanding sometimes difficult. A substantial part of my comments is on the English (see below in detail). Interestingly, Introduction, M&M and (partially) Results are exempt from this and seem to have been written by a different person than other parts, especially Discussion. I therefore recommend this manuscript to be considered for publication only after revision. This includes a check of the English by a competent person.